# Genome-Wide Association Study (GWAS) Analysis of Camelina Seedling Germination under Salt Stress Condition

**Zinan Luo, Aaron Szczepanek and Hussein Abdel-Haleem \***

US Arid Land Agricultural Research Center, USDA ARS, Maricopa, AZ 85138, USA;
Lily.Luo@USDA.GOV (Z.L.); Aaron.Szczepanek@USDA.GOV (A.S.)
\* Correspondence: Hussein.Abdel-Haleem@USDA.GOV

**Abstract:** *Camelina sativa* is an important renewable oilseed crop for biofuel and feedstock that can relieve the reliance on petroleum-derived oils and reduce greenhouse gases and waste solids resulting from petroleum-derived oils consumption. *C. sativa* has recently seen revived attention due to its high oil content, high omega-3 unsaturated fatty acids, short life cycle, broader regional adaptation, and low-input agronomic requirements. However, abiotic stress such as salinity stress has imposed threatens on plant photosynthesis and growth by reducing water availability or osmotic stress, ion ($Na^+$ and $Cl^-$) toxicity, nutritional disorders and oxidative stress yield. There still remains much to know for the molecular mechanisms underlying these effects. In this study, a preliminary study applying 10 *C. sativa* cultivars to be treated under a gradient NaCl concentrations ranging from 0–250 mM and found that 100 mM was the optimal NaCl concentration to effectively differentiate phenotypic performance among different genotypes. Then, a spring panel consisting of 211 *C. sativa* accessions were germinated under 100 mM NaCl concentration. Six seedling germination traits, including germination rate at two stages (5-day and 9-day seedling stages), germination index, dry and fresh weight, and dry/fresh ratio, were measured. Significant correlations were found between the germination rate at two stages as well as plant biomass traits. Combining the phenotypic data and previously obtained genotypic data, a total of 17 significant trait-associated single nucleotide polymorphisms (SNPs) for the germination rate at the two stages and dry weight were identified from genome-wide association analysis (GWAS). These SNPs are located on putative candidate genes controlling plant root development by synergistically mediating phosphate metabolism, signal transduction and cell membrane activities. These identified SNPs could provide a foundation for future molecular breeding efforts aimed at improved salt tolerance in *C. sativa*.

**Keywords:** *Camelina sativa*; salt stress; germination; GWAS; phosphate metabolism; signal transduction; cell membrane activity

## 1. Introduction

*Camelina sativa* L. Crantz belongs to the Brassicaceae (Cruciferae) family and is an economic crop originating from Southeastern Europe and Southwestern Asia [1,2]. Due to its high level of omega-3 essential fatty acids, favorable agronomic characteristics, and potential to be a biofuel resource, *C. sativa* has revived researchers' interests in recent years after being replaced by canola in the 1950s [3]. Over 90% of the total oil content in *C. sativa* is unsaturated fatty acids, of which omega-3 $\alpha$-linolenic essential fatty acid accounts for up to 40% [3]. In addition, advantageous agronomic characteristics such as low-input requirements for water, nutrients and pesticides, early maturity and resistance against insects and diseases [1,2,4], enable the growth of *C. sativa* in marginal regions without employing

arable land for food production. These oil quality characteristics, together with the agronomic benefits, make *C. sativa* an ideal and viable resource for both biofuel and animal feedstock.

Regardless of these potentials, inadequate breeding efforts carried out on *C. sativa* so far have mainly focused on improving seed yield and/or certain oil compositions without paying attention to both biotic and abiotic stress resistance [5]. Moreover, the low genetic diversity in the currently available released germplasm limits breeding progress [5]. In order to extend *C. sativa's* acreage and production into marginal lands and to develop a sustainable agricultural system, it is important to breed for advanced germplasm with enhanced productivity and resistance to abiotic stresses.

In general, salinity stress is one of the most harmful abiotic stresses affecting plant growth. Crop production in approximately 20% of the world's arable lands and nearly half of irrigated lands is adversely affected by salinity stress [6,7]. One of the effects of salinity stress is the inhibition of plant photosynthesis and growth by reducing water availability or osmotic stress, ion ($Na^+$ and $Cl^-$) toxicity, nutritional disorders and oxidative stress [7,8]. Previous studies suggested that different plant growth stages might show different levels of salt tolerance [9,10]. Plants may be tolerant to salt stress at one growth stage but sensitive at another, and the effects of salt tolerance at different plant growth stages are usually independent of each other [9,10]. Researchers noted that selection for salt tolerance at germination, seedling stage or early vegetative growth stage did not improve tolerance in adult plants [11,12], while in other studies, salinity stress at early seedling development stage affected plant growth at later growing stages [13] by reducing canopy stand [14], vegetative growth, grain yield and protein content [15]. Therefore, as for plant species with few studies regarding salinity tolerance such as *Camelina sativa*, investigating the effects of salinity stress at seedling germination stage is the first step for further research exploring effects on plant growth, vegetative development and seed yield.

While plant species may respond in different ways and mechanisms to abiotic stresses, many stresses share common general responses [8]. For example, *C. sativa* is somewhat resistant to drought stress by developing deep root systems or mediating metabolic and signaling pathways, which are common features of both drought and salt tolerance [16,17]. Salt stress condition affects root elongation and root architecture in plants, reducing cell size, cell division and altering differentiation patterns [8,17]. Mineral metabolism such as nitrogen, phosphorus, and sulfur nutrition are also affected by salinity stress [18,19]. In addition, complicated plant hormone signaling transduction systems and the corresponding cell activities are also involved in mitigating detrimental effects on plant growth and development under salt stress. Therefore, in order to improve salt tolerance in *C. sativa*, a global genetic dissection of early plant growth traits on a genome-wide level is necessary to unravel the candidate genes related to plant growth and development under salt stress.

The rapid development of next-generation sequencing (NGS) technologies enables the utilization of marker-assisted selection (MAS) as a tool to accelerate genetic gains in a more cost-effective way. Genome-wide association studies (GWAS) could overcome the limitations of traditional quantitative trait loci (QTL) mapping and can be used to dissect minor genetic factors underlying complex traits in crop species [20]. The identified QTLs can be used in MAS to improve genetic gains in a more efficient manner. To date, GWAS has been used to identify candidate genes affecting plant growth and development under salt stress in rice [21], barley [22], soybean [23], sesame [24], and alfalfa [25]. In the current study, we hypothesized that the significant trait-associated single nucleotide polymorphisms (SNPs) could be directly or indirectly related to plant germination and growth in response to salinity stress. We combined the phenotypic data and previously published genotypic data [26] to conduct a GWAS analysis and identified significant trait-associated SNPs and candidate alleles/genes that can putatively regulate molecular mechanisms, such as phosphate metabolism, signal transduction and cell membrane activities, under salt tolerance at the early seedling growth stage in the spring panel of *C. sativa*, which consisted of accessions originally from different geographical regions. This study could lay a foundation in future breeding efforts to

improve salt tolerance in *C. sativa* via mediating mineral metabolism, signaling transductions and cell response activities.

## 2. Materials and Methods

### 2.1. Plant Materials and Salt Stress Tolerance Assays

To determine the optimal salt concentration to detect the phenotypic variation to salt stress in the entire spring panel, ten *C. sativa* cultivars were germinated at NaCl concentrations ranging from 0, 25, 50, 75, 100, 150, 200, up to 250 mM. The ten tested cultivars include Calena, Robinson, Galena, Blaine Creek, Pronghorn, Celine, 13CS0787-8, CO46, Midas and Suneson. Seeds of each cultivar were surface sterilized with bleach for 5 min, followed by washing for 30 s with 70% ethanol and 3–4 rinses with sterilized distilled water. Fifteen seeds of each cultivar were then left on blue blotter paper (Hoffman Manufacturing, Inc, Corvallis, OR, USA) for germination in Petri dishes containing 10 mL of a NaCl solution of different concentrations. All Petri dishes were placed in controlled room temperature under 12 hr light with white fluorescent light providing an intensity of 200 $\mu$mol s$^{-1}$ m$^{-2}$, in randomized complete block design (RCBD) with factorial arrangements experiment. Each treatment was replicated three times. Germination percentages were recorded after 11 days.

Based on preliminary experiment results, the optimal salt concentration, 100 mM, was used to differentiate salt stress responses in the spring camelina diversity panel (Luo et al., 2019). Twenty-five seeds of each of the 211 *C. sativa* accessions were surface sterilized and germinated following the same conditions of the preliminary experiment. *C. sativa* accessions were germinated in an alpha-lattice design with three replications. Cultivars Celine and Galena were included in each block as checks. Six traits measured during the experiment included: germination rate after 5 (germone) and 9 days (germtwo) of planting, GI (germination index, the sum of germinated seedlings/day from the first and second stages), fresh weight (single seedling fresh weight, mg), dry weight (single seedling dry weight, mg), and dry/fresh ratio (% of dry/fresh weight). Both fresh and dry weight were determined for 9-day old seedlings.

### 2.2. Phenotypic Analysis

To determine the optimum NaCl salt concentration, the statistical model was calculated using the general linear model procedure (PROC GLM) of SAS software as $Y_{ijk} = \mu + B_k + \tau_i + \beta_j + \tau\beta_{ij} + \varepsilon_{ijk}$, where $\mu$ is the mean, $\tau i$ is the effect of salt concentration factor *i*, $\beta j$ is the effect of camelina genotype factor j, $\tau\beta ij$ is interaction effects of combination ij and $\varepsilon_{ijk}$ is the effect of experimental error of salt concentration *i* on camelina genotype *j* and block *k*. To compare among salt concentrations and camelina genotypes Least Significant Differences (LSD) were calculated at the significance level ($p = 0.05$).

Salt tolerance related traits of the whole camelina panel were analyzed using mixed linear models (MLM) procedure (PROC MIXED) of SAS software: The statistical model was calculated as: $Y_{ijk} = \mu + G_i + R_j + B(R)_{jk} + \varepsilon_{ijk}$, where $\mu$ is the mean, $G_i$ is the effect of accession *i*, $R_j$ is the effect of replicate *j*, $B(R)_{jk}$ is the effect of block *k* nested in the replicate *j* and $\varepsilon_{ijk}$ is the effect of experimental error of accession *i* in replication *j* and block *k*. The effects of replication and accessions were considered as random effects (Statistical Analysis System, SAS Institute, 2001). Based on MLM, best linear unbiased predictors (BLUPs) were calculated for each trait. Broad-sense heritability on an entry-mean basis was calculated as $h^2 = (\sigma^2_{accesion}/(\sigma^2_{accesion} + \sigma^2_e/r)$ [27,28], where $\sigma^2_{accesion}$ is the genetic variance among accessions, $\sigma^2_e$ is the variance of experimental error, and r is the number of replications. Correlation coefficients among six different traits were calculated by Pearson's method at a significance level of $p < 0.05$ using the "Corrplot" package implemented in R [29].

### 2.3. Genotypic Data, Population Genetics, Linkage Disequilibrium (LD)

DNA extraction, Genotyping-by-sequencing (GBS) and sequencing data analyses of *C. sativa* spring panel were summarized by Luo et al. [26]. Briefly, the SNPs with missing data > 20% and MAF < 0.05 were removed for the subsequent analyses [30]. The GBS of 211 *C. sativa* accessions resulted in 6,192 high-quality SNPs that were physically mapped across 20 chromosomes [26]. The resulting SNPs in VCF file were converted to HAPMAP format using TASSEL [31]. Detailed procedures of population genetics analyses using STRUCTURE v2.3.4 [32] were provided by Luo et al. [20]. Population genetics analyses identified two subpopulations based on the results from Structure Harvester [33]. This result was in accordance with the principle component analysis (PCA) based on the geographical separation of 211 *C. sativa* accessions [20]. Linkage disequilibrium (LD) between SNPs on each chromosome and on an overall level was estimated using $r^2$ from TASSEL5.0 [31] as summarized by Luo et al. [20]. Based on the $r^2$ obtained from LD analyses, 3417 SNP pairs with strong LD ($r^2$ value larger than 0.6) were filtered, which was used to identify SNPs in strong LD with the significant trait-associated SNPs.

### 2.4. Genome Wide Association Study (GWAS) Analyses

Genome-wide association studies (GWAS) were conducted using FarmCPU (Fixed and random model Circulating Probability Unification) [34], which is a newly proposed method implemented in rMVP package in R version 3.6.3 [35]. In order to effectively control false positives and reduce false negatives, the method iteratively performs marker tests with pseudo-quantitative trait nucleotides (QTNs) as covariates in a fixed-effect model and keeps optimizing pseudo-QTNs in a random-effect model until no new pseudo QTNs are added [34]. The capability of removing confounding effects between testing markers and kinship, preventing model overfitting and controlling false positives enable FarmCPU, has become one of the most efficient methods in current GWAS [36]. In this study, best linear unbiased predictions (BLUPs) of six seedling germination traits collected from the 211 *C. sativa* accessions and 6192 numerically SNPs [26] were used to perform FarmCPU algorithm in R studio. Kinship (K) and PCA (P) were calculated using rMVP package. The first five principal components (PCs) were included in the GWAS model to correct for hidden population structure. The maxLoop value was set to 100, and the method was set as "FaST-LMM". The threshold for p-value selection was set to 1.0 divided by the number of SNPs for all traits. Subsequently, p-values of significant marker–trait associations were adjusted using a false discovery rate analysis (FDR).

### 2.5. Candidate Gene Identification

*In silico* mapping of SNPs was performed using the *C. sativa* genome browser (https://www.ncbi.nlm.nih.gov/genome/gdv/browser/?context=genome&acc=GCF_000633955.1) provided by Agriculture and AgriFood Canada. The LD blocks harboring significant trait-associated markers and their flanking markers with strong LD ($r^2 > 0.6$) were defined as candidate gene regions as suggested by Knoch et al. [36]. All genes within the LD blocks were considered for candidate genes and annotated for predictive functions using *C. sativa* genome annotation database incorporated in snpEff [37]. For significant trait-associated markers outside the coding regions, the 2.0-kb flanking regions on either side were searched for candidate genes [20].

## 3. Results and Discussion

### 3.1. Phenotypic Diversity in Seedling Germination Traits under Salt Stress

Analyses of variance (ANOVAs) of the preliminary salt stress experiment revealed significant effects of different NaCl concentrations ($p < 0.0001$), genotypes ($p < 0.006$) and their interaction ($p < 0.0001$) on seed germination of the studied camelina cultivars (Table 1). The NaCl concentrations of 75 mM and 100 mM were significantly different from other low concentrations (25 and 50 mM) and high concentrations (150, 200 and 250 mM) (Figure 1a). Even though both the effects of 75 mM and 100 mM NaCl were significantly different from others, 75 mM NaCl hardly differentiated salt stress

responses among the 10 different *C. sativa* cultivars (Figure 1b), while large phenotypic variations were observed under 100 mM NaCl, meaning that this concentration can easily distinguish salt-tolerant cultivars from salt-sensitive ones (Figure 1b). Under 100 mM NaCl, the germination rate of different cultivars ranged from 0% in 13CS0787-8 to 60% in Suneson (Figure 1b). Based on these results, 100 mM NaCl proved to be the optimal concentration to induce a wide range of phenotypic variations among tested genotypes, this concentration was further used to screen the entire spring panel.

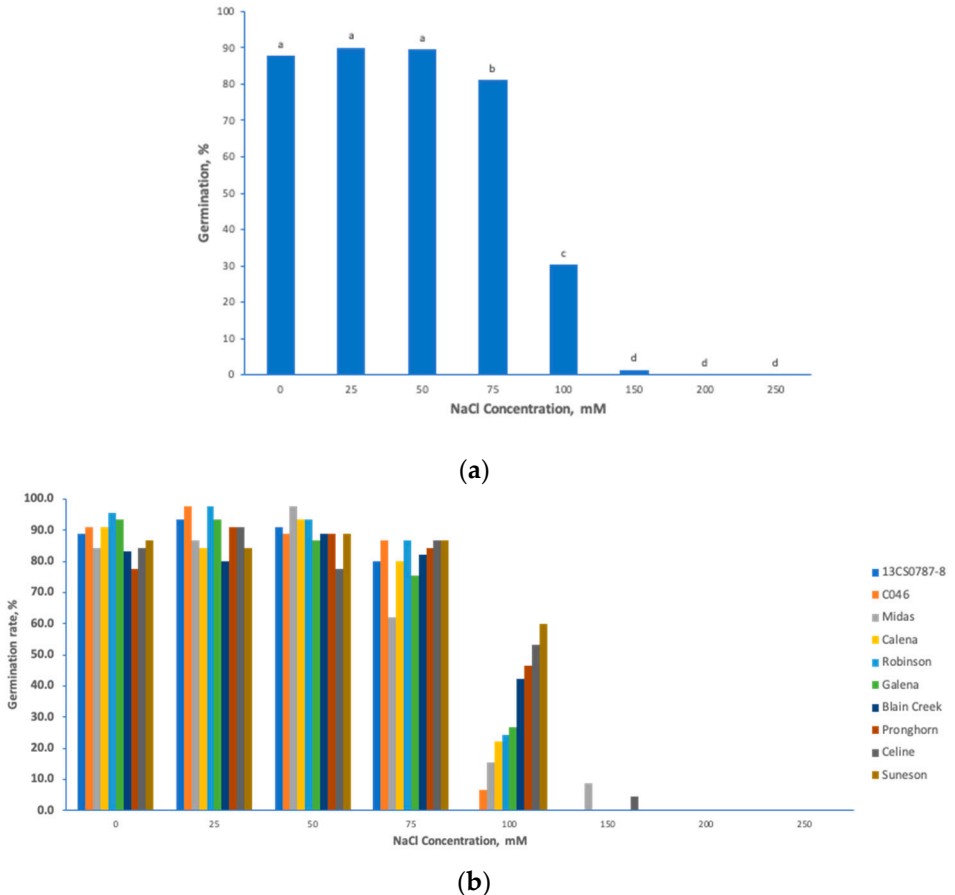

**Figure 1.** Preliminary salt treatment study determines the optimal salt stress conditions for *Camelina sativa*. (**a**) germination percentage of *Camelina sativa* cultivars seedlings under different NaCl salt concentrations; and (**b**) germination percentage of 10 *C. sativa* cultivars seedling under different NaCl salt concentrations. Note: letters "a" or "b" refer to significant difference levels.

**Table 1.** Two-way Analysis of Variance (ANOVA) of germination percentage.

| Source of Variance | Df | Mean Square |
|---|---|---|
| Replication | 2 | 0.83 $^{ns}$ |
| NaCl Concentration (Conc) | 7 | 495.33 *** |
| Cultivar (Gen) | 9 | 1.43 ** |
| Conc*Gen | 63 | 1.93 *** |

Note: $^{ns}$ is none significant, *** is significant at < 0.0001, and ** is significant at < 0.01.

Following the same experimental conditions, the 211 *C. sativa* accessions were germinated under 100 mM NaCl, and six seedling germination traits were recorded. The statistical summary of all the six traits and the corresponding broad-sense heritability are shown in Table 2. Pearson's pairwise correlation analysis (Table 3) among six traits showed that germination percentage after 5 (germone) and 9 days (germtwo) and GI were significantly correlated with each other (r = 0.9277; *p* < 0.0001), indicating that either one of these three traits can be used to estimate seed germination under salt

stress. Figure 2 showed a wide phenotypic variation among the 211 accessions in terms of germination rate at two stages, GI and dry weight. During the seedling germination stage, the most salt-sensitive accessions were German accession (Cs118) and Russian accession (Cs223) with only 25% and 27% of their seeds germinated respectively after 5 days under 100 mM NaCl (Figure 2a). On the other hand, the most tolerant accessions were the Russian accession (Cs146) with a 94% germination rate, followed by German accessions (Cs172, Ca001 and Cs163), Bulgarian accession (Cs009) and Ukrainian accession (Cs189) with 93% of seeds germinated (Figure 2a). German accessions (Cs002, Cs100, Cs101 and Cs234) and Kyrgyz accession (Cs003) ranked the highest in seedling dry weight under stress, while the lowest were Georgian accessions (Cs195 and Cs196) and Russian accessions (Cs038 and Cs043) (Figure 2b). These accessions can be selected as good candidates for salt-tolerant parents in future breeding programs. In addition, all six traits showed high broad-sense heritability (Table 2), indicating that the possibility of improving the genetic gains of salt tolerance at the germination stage is effective.

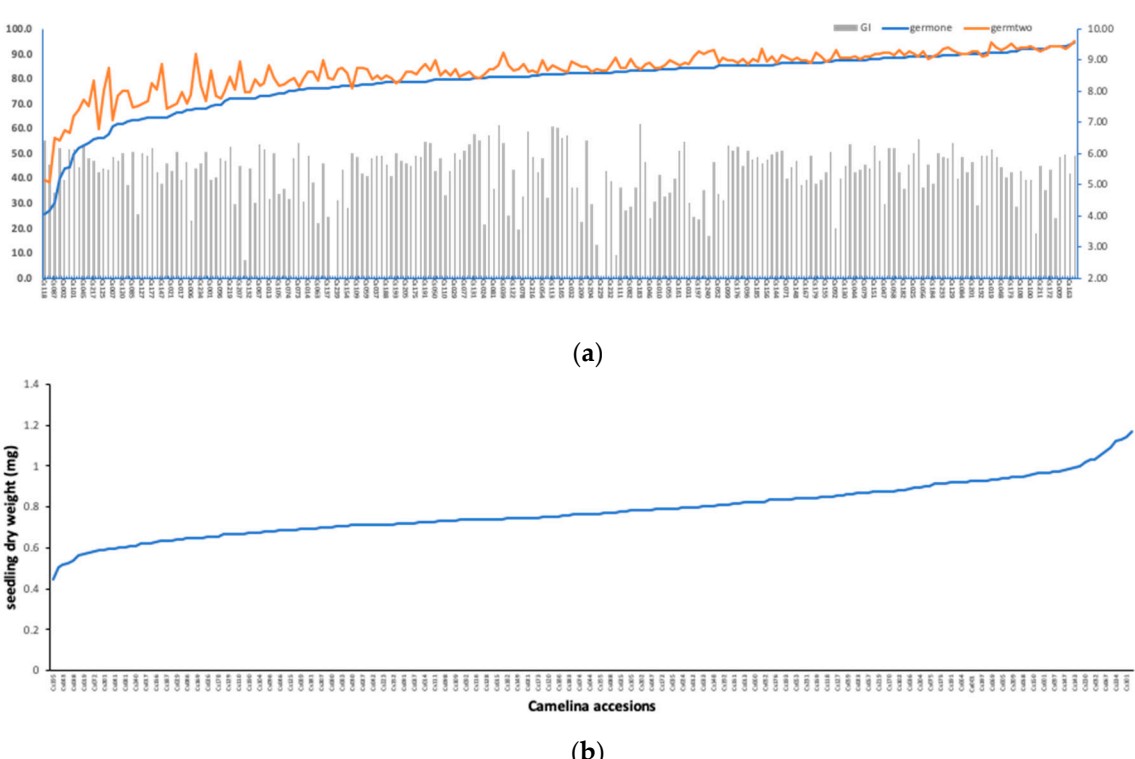

**Figure 2.** (**a**) Best linear unbiased predictor (BLUP) values of germination rate after 5- and 9-days of planting and germination index (GI) for 211 *C. sativa* accessions germinated under 100 mM NaCl; and (**b**) dry weight BLUPs of the 211 *C. sativa* accessions germinated under 100 mM NaCl.

**Table 2.** Statistics summary of germination traits in Camelina sativa under salt stress condition (100 mM NaCl).

| Variable | Mean | Std Dev | Minimum | Maximum | Heritability (*H*) |
|---|---|---|---|---|---|
| Germone (%) | 78.3136 | 17.0766 | 11.76 | 100.00 | 0.8303 |
| Germtwo (%) | 82.6463 | 14.1230 | 16.00 | 100.00 | 0.7625 |
| GI | 5.3514 | 1.2363 | 0.67 | 7.50 | 0.8615 |
| Fresh weight (mg) | 18.8544 | 5.9687 | 5.16 | 53.98 | 0.7752 |
| Dry weight (mg) | 0.7835 | 0.1698 | 0.12 | 1.47 | 0.8834 |
| Dry/fresh ratio (%) | 4.4482 | 1.4694 | 0.62 | 15.43 | 0.8140 |

Note: Germone represents germination rate at 5-day seedling stage; germtwo represents germination rate at 9-rate seedling stage; GI represents germination index.

**Table 3.** Pearson's correlation among six different traits under salt stress for 211 *Camelina sativa* accessions.

|  | Germone | Germtwo | GI | Fresh | Dry | Dry/Fresh Ratio |
|---|---|---|---|---|---|---|
| Germone | 1 | 0.9277 | 0.8682 | 0.2899 | −0.1275 | −0.5147 |
| Germtwo | <0.0001 | 1 | 0.8455 | 0.2507 | −0.0539 | −0.4027 |
| GI | <0.0001 | <0.0001 | 1 | 0.1424 | −0.2604 | −0.4490 |
| Fresh | <0.0001 | 0.0002 | 0.0388 | 1 | 0.5525 | −0.6502 |
| Dry | 0.0644 | 0.4360 | 0.0001 | <0.0001 | 1 | 0.1644 |
| Dry/fresh ratio | <0.0001 | <0.0001 | <0.0001 | <0.0001 | 0.0168 | 1 |

Note: Germone represents germination rate at 5-day seedling stage; germtwo represents germination rate at 9-day seedling stage; GI represents germination index.

### 3.2. GWAS Analysis of Seedling Germination Traits under Salt Stress

A total of 17 SNP markers were observed to be significantly associated with three out of six traits: germination rates at 5- and 9-day stage and seedling dry weight (Table 4, Figure 3). A comprehensive list of all significant trait-associated markers and the corresponding candidate genes was provided in Table 4 and Figure 3. Two SNP markers located on chromosome 7 and 20 were associated with both germination rates at 5-day and 9-day seedling development stages.

The SNP located on chromosome 7 (S7_2473684) is significantly associated with germination rates at 5- and 9-day traits (Table 4, Figure 3), and represents the gene encoding eIF2$\alpha$ kinase GCN-2 like protein, which is the only known kinase in plants that phosphorylates the $\alpha$-subunit of eukaryotic initiation factor 2 (eIF2$\alpha$) [38,39]. Phosphorylation of eIF2$\alpha$ is one of several mechanisms regulating global translation. When unphosphorylated, eIF2 that is bound to guanosine triphosphate (GTP) initiates mRNA translation by delivering the initiator methionyl-tRNA to the small ribosomal subunit (40S) [39,40]. Once phosphorylated by kinases, eIF2$\alpha$ then becomes eIF2B, a poisoned substrate of the guanine nucleotide exchange factor, repressing global translation [39,41]. GCN2 is one of the kinases that phosphorylates eIF2$\alpha$ and can be activated under cold and salt stress, resulting in reduced primary root growth in *Arabidopsis thaliana* [39]. Therefore, the SNP (S7_2473684) located on the gene encoding eIF2 kinase GCN-2 like protein might be also related to root germination regulation in *C. sativa* under salt stress.

Another SNP (S20_15783417) was also found to be significantly associated with both germination rates at two stages (Table 4, Figure 3). This SNP is located on a gene encoding dextranase, which is a marine enzyme usually secreted by bacteria, showing high salt tolerance [42]. In plants, dextranase enzyme function could be related to the breakdown of dextran (a long chain sugar molecule) in sugar manufacture [43]. Dextran is thought to be induced by salt stress due to its function in bulk-flow membrane endocytosis in *A. thaliana* [44]. However, whether the dextranase activities during the germination stage are directly or indirectly triggered by increased dextran synthesis in plant roots or by increased root bacteria attachment triggered by salt stress needs more investigation.

Our study identified SNPs significantly associated with dry weight under salt stress, and candidate gene functions were annotated. One SNP located on chromosome 1 (S1_1910174), control expression of serine/threonine-protein phosphatase PP1 isozyme 9, was identified to be significantly associated with dry weight affected by salt stress (Table 4, Figure 3). Previous studies reported a negative relationship between salinity level and phosphorus metabolism in plant roots and/or shoots [7,45,46]. However, an increase in phosphatase and ATPase activities have been observed in both roots and shoots under salt stress [19,45]. The protein phosphatases can be divided into four families based on primary sequence and catalytic mechanisms: phosphoprotein phosphatase (PPP), the $Mg^{2+}$- or $Mn^{2+}$-dependent protein phosphatase (PPM)/protein phosphatase 2C (PP2C), the phosphotyrosine phosphatase (PTP), and the aspartate (Asp)-dependent enzyme families [47]. Seven subfamilies that are sequence-related and share identical catalytic mechanisms have been categorized within the PPP family, these include Ser/Thr protein phosphatase type one (PP1), PP2 (PP2A), PP3 (PP2B), PP4, PP5, PP6, and PP7 [47].

Some of the above enzymes such as PP1, PP2A, PP2B have been found to be responsive to salt stress conditions in plant species [48–53]. Novel members of PPP family have also been characterized, including the protein phosphatases with Kelch-like repeat domains (PPKL) and Shewanella-like protein (SLP) phosphatases [47,54].

Four PPKLs were characterized in *A. thaliana*: BSU1, BSL1, BSL2, and BSL3 [55–57], which showed a functional relevance in interactively affecting brassinosteroid signaling [54]. Brassinosteroid signaling was found to mediate plant development and adaptation under salt stress [56,58]. An SNP located on chromosome 3 (S3_3511428) annotated as BSL2 was significantly associated germination rate at the 9-day seedling stage (Table 4, Figure 3), although no direct evidence showed that the increased activity of BSL2 was triggered by salt stress. In addition, the SNP S8_24097673 on chromosome 8 was associated with dry weight under salt stress and located in the gene encoding transcription factor B1M1-like (Table 4) also showed the potential indirect effect responding to salt stress since BIM1 is a bHLH protein involved in brassinosteroid signaling [59].

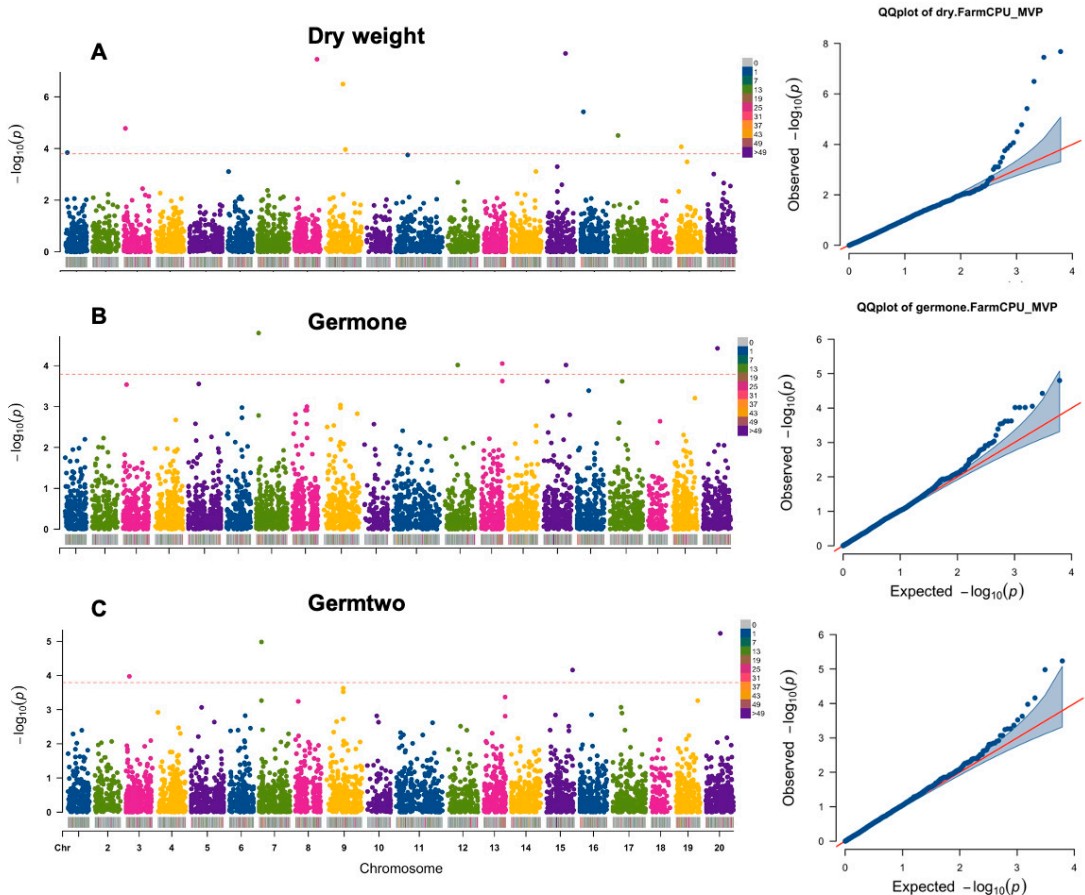

**Figure 3.** Manhattan plots for significant trait-associated single nucleotide polymorphisms (SNPs) in Camelina sativa under salt stress. (**A**) dry weight; (**B**) germination percentage after 5 days on 100 mM NaCl; (**C**) germination percentage after 9 days on 100 mM NaCl.

The SNP S19_4398468 (Table 4, Figure 3) was annotated as the gene encoding exocyst complex component SEC8, which is a single subunit at the site of active exocytosis responsible for intracellular transport between Golgi apparatus and plasma membrane [60]. Although there was no direct evidence indicating the relationship between SEC8 and salt tolerance, the exocyst subunits showed high homology with EXO70 family in *A. thaliana* [61]. EXO70 family proteins were reported to be responsible for vesicular trafficking, regulation of actin polarity and vesicle transportation [62]. Some genes in this family were found to be upregulated by salt stress [63], but the exact function of the exocyst complex

in plants against abiotic stress is not clear. Our study might indicate the indirect relationship between exocyst complex SEC8 and salt stress tolerance in *C. sativa*.

## 4. Conclusions

The present study determined the optimal salt concentration (100 mM) to differentiate camelina accessions in response to salt stress at the seedling germination stage. The GWAS analysis identified 17 significant trait-associated SNP markers that were putatively related to the germination rate at two stages (5-day and 9-day after treatment) and dry weight biomass under salinity stress. Two SNPs were associated with the variation in germination rate at both stages, which showed significant Pearson's correlation. The putative candidate genes that indirectly affect salt tolerance in *C. sativa* might be related to the mediation of phosphate metabolism, signal transductions and cell membrane activities. These identified SNPs, together with a high heritability of seedling germination traits, would provide insight into the expectation of promising genetic gains in future molecular breeding efforts aimed at improving salt tolerance in *C. sativa*. However, this study was based on a growth chamber experiment; further larger-scale multilocation field trials in successive multiple years will be needed to study the Genotype x Environment (GxE) interaction and to validate those identified SNPs as well as the associated candidate genes.

**Table 4.** Annotation of genes harboring significant trait-associated SNPs across 20 chromosomes in *Camelina sativa* under salt stress.

| SNP | Chr | Position | Effect | *p*-Value | FDR adj *p*-Value | Trait | Nearest Gene | Gene Annotation | Protein Function |
|---|---|---|---|---|---|---|---|---|---|
| S1_1910174 | 1 | 1910174 | 0.0259 | 0.00014389 | 0.00014389 | Dry weight | LOC104713675 | serine/threonine-protein phosphatase PP1 isozyme 9 | phosphatase activity toward para-nitrophenyl phosphate (pNPP) in vitro |
| S3_1462975 | 3 | 1462975 | −0.0537 | $1.67 \times 10^{-5}$ | $3.97 \times 10^{-5}$ | Dry weight | LOC104761613 | pre-mRNA-processing factor 39 | pre-mRNA splicing |
| S3_3511428 | 3 | 3511428 | 5.8912 | 0.00010599 | 0.00011582 | Germtwo | LOC104766624 | serine/threonine-protein phosphatase BSL2 | Phosphatase involved in elongation process, probably by acting as a regulator of brassinolide signaling |
| S7_2473684 | 7 | 2473684 | 4.9858 | $1.59 \times 10^{-5}$ | $3.97 \times 10^{-5}$ | Germone | LOC104699973 | eIF-2-alpha kinase GCN2-like | Metabolic-stress sensing protein kinase that phosphorylates the alpha subunit of eukaryotic translation initiation factor 2 (eIF-2-alpha/EIF2S1) on "Ser-52" in response to low amino acid availability |
| S7_2473684 | 7 | 2473684 | 3.6988 | $1.04 \times 10^{-5}$ | $3.29 \times 10^{-5}$ | Germtwo | LOC104699973 | eIF-2-alpha kinase GCN2-like | Metabolic-stress sensing protein kinase that phosphorylates the alpha subunit of eukaryotic translation initiation factor 2 (eIF-2-alpha/EIF2S1) on "Ser-52" in response to low amino acid availability |
| S8_24097673 | 8 | 24097673 | 0.035 | $3.53 \times 10^{-8}$ | $3.35 \times 10^{-7}$ | Dry weight | LOC104708282 | transcription factor BIM1-like | DNA binding; protein dimerization activity |
| S9_18205018 | 9 | 18205018 | 0.0484 | $3.20 \times 10^{-7}$ | $2.03 \times 10^{-6}$ | Dry weight | NA | NA | NA |
| S9_20790122 | 9 | 20790122 | 0.0225 | 0.00010973 | 0.00011582 | Dry weight | LOC104715504 | NA | NA |
| S12_13693594 | 12 | 13693594 | 47.5695 | $9.62 \times 10^{-5}$ | 0.00011424 | Germone | LOC104731617 | inactive disease resistance protein RPS4 | Resistance to Pseudomonas syringae 4 |
| S12_13693600 | 12 | 13693600 | 47.5695 | $9.62 \times 10^{-5}$ | 0.00011424 | Germone | LOC104731617 | inactive disease resistance protein RPS4 | Resistance to Pseudomonas syringae 4 |
| S13_23380011 | 13 | 23380011 | −5.1802 | $8.85 \times 10^{-5}$ | 0.00011424 | Germone | NA | NA | NA |
| S15_20134089 | 15 | 20134089 | −0.051 | $2.10 \times 10^{-8}$ | $3.35 \times 10^{-7}$ | Dry weight | NA | NA | NA |
| S15_24396991 | 15 | 24396991 | 47.5695 | $9.62 \times 10^{-5}$ | 0.00011424 | Germone | LOC104748747 | DNA replication licensing factor MCM4-like | DNA replication initiation and elongation |
| S15_28433235 | 15 | 28433235 | −4.4191 | $6.94 \times 10^{-5}$ | 0.00011424 | Germtwo | NA | NA | NA |
| S16_2915367 | 16 | 2915367 | 0.0687 | $3.81 \times 10^{-6}$ | $1.81 \times 10^{-5}$ | Dry weight | LOC104749495 | DEAD-box ATP-dependent RNA helicase 21-like | mRNA splicing |
| S17_4964987 | 17 | 4964987 | −0.0395 | $3.14 \times 10^{-5}$ | $6.63 \times 10^{-5}$ | Dry weight | LOC104755363 | NA | NA |
| S19_4398468 | 19 | 4398468 | −0.0386 | $8.57 \times 10^{-5}$ | 0.00011424 | Dry weight | LOC104764745 | exocyst complex component SEC8-like | vesicular trafficking, regulation of actin polarity and vesicle transportation |
| S20_15783417 | 20 | 15783417 | −9.7872 | $3.75 \times 10^{-5}$ | $7.13 \times 10^{-5}$ | Germone | LOC104772556 | dextranase-like | catalyze Endohydrolysis of (1->6)-alpha-D-glucosidic linkages in dextran |
| S20_15783417 | 20 | 15783417 | −7.7913 | $5.84 \times 10^{-6}$ | $2.22 \times 10^{-5}$ | Germtwo | LOC104772556 | dextranase-like | catalyze Endohydrolysis of (1->6)-alpha-D-glucosidic linkages in dextran |

Note: Germone represents germination rate at 5-day seedling stage; germtwo represents germination rate at 9-day seedling stage; GI represents germination index.

**Author Contributions:** Conceptualization, H.A.-H.; methodology, A.S., Z.L.; software, H.A.-H. and Z.L.; validation, H.A.-H., A.S. and Z.L.; formal analysis, H.A.-H. and Z.L.; investigation, H.A.-H, A.S. and Z.L.; resources, H.A.-H.; data curation, H.A.-H., A.S.; writing—original draft preparation, Z.L.; writing—review and editing, H.A.-H, and Z.L.; visualization, H.A.-H., and Z.L.; supervision, H.A.-H.; project administration, H.A.-H.; funding acquisition, H.A.-H. All authors have read and agreed to the published version of the manuscript

**Funding:** This research was funded by the United States Department of Agriculture-Agricultural Research Service (USDA-ARS) 2020-21410-007-00D and National Institute of Food and Agriculture (NIFA) Grant 2016-67009-25639.

**Acknowledgments:** We would like to thank Amber Dearstyne, Harmony Glover and Avery Luna for the assistance with data collection and organization. Mention of trade names or commercial products in this publication is solely for providing specific information and does not imply recommendation or endorsement by the United States Department of Agriculture. The USDA is an equal opportunity provider and employer.

**Conflicts of Interest:** The authors declare no conflict of interest.

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
