# Peer review of "Genome-Wide Association Study (GWAS) Analysis of Camelina Seedling Germination under Salt Stress Condition"

_agronomy, doi:10.3390/agronomy10091444_

Round 1

Reviewer 1 Report

The manuscript by Luo et al. entitled “Genome-wide association study (GWAS) analysis of 2 camelina seedling germination under salt stress 3 condition” reports six seedling germination traits of Camelina sativa could associate with 17 SNPs that position on plant root development related genes. The work is a technically sound piece of research, and well within the scope of Agronomy. It is my sincerest hope that the authors may check the expression level of the deduced genes using the seedlings of C. sativa varieties (for examples, mRNA-processing and other transcriptional factors), to further demonstrate the correlations between the SNPs and the germination traits under salt stress conditions. The authors may also analyze the relationships between the other traits and the SNPs base on the obtained data to provide another perspective to evaluate the potential agricultural applications.

Problem need to be addressed:

  1. Detailed seedling germination conditions, for example, the light source and density
  2. Check the spellings

Author Response

Dear Reviewer:

Zinan

Reviewer 2 Report

Current manuscript "genome-wide association study (GWAS) analysis of camelina seedling germination under salt stress condition authored by Luo et al. has interesting results. However, number of improvements are required before accepting this paper.

Abstract:

  1. The objective of study is not clearly mentioned in abstract. Please included the objective
  2. Please mention all the treatments of salinity in the abstract  

Introduction:

  1. Page 2 line 45. Please change the statement “salinity is the most harmful” to “salinity is one of the most harmful”
  2. No hypothesis is mentioned in last paragraph of the introduction. Please state a clear hypothesis.

Materials and methods

  1. Please incorporate the changes highlighted in yellow.
  2. Please mention parentage and pedigree of the approved cultivars and areas of collection for different accessions in the supplementary table if possible.
  3. No information or reference is provided about the sampling and genotyping of the accessions
  4. Nothing is mentioned about second years’ data as in my opinion one-year data is not enough to generalize GWAS studies
  5. Please include two-way analysis of variance to indicate significance level and their interaction.

Results and discussion

  1. Please incorporate the changes highlighted in yellow.
  2. Please reanalyze the data by two factor ANOVA or two-way analysis of variance and provide short table. For help you can see the manuscript “Shokat, S; Sehgal, D; Vikram, P; Liu, F; Singh, S; 2020. Molecular markers associated with agro-physiological traits under terminal drought conditions in bread wheat. International Journal of Molecular Sciences. 21(9): 3156.”
  3. Detail of results of section is provided but not properly discussed in section 3.1. Please discuss the results.
  4. Section 3.2 can be moved to materials and methods section
  5. Please delete the paragraph from line 247-251. It seems unnecessary.

Conclusion

  1. Please indicate in the conclusion that further experimentation is required to validate these results as the this is only one year data

Author Response

Dear reviewer 2:

Please see the attachment. Thank you for your time and effort in reviewing my MS. 

Zinan
